# Controlled Release of Thymol by Cyclodextrin Metal-Organic Frameworks for Preservation of Cherry Tomatoes

**DOI:** 10.3390/foods11233818

**Published:** 2022-11-26

**Authors:** Zhicheng Li, Yanan Sun, Xiaodan Pan, Tong Gao, Ting He, Chun Chen, Bin Zhang, Xiong Fu, Qiang Huang

**Affiliations:** 1Guangzhou Restaurant Group, Likofu Food Company Ltd., Guangzhou 511445, China; 2Guangdong Province Key Laboratory for Green Processing of Natural Products and Product Safety, School of Food Science and Engineering, South China University of Technology, Guangzhou 510640, China; 3China-Singapore International Joint Research Institute, Guangzhou 511363, China

**Keywords:** CD-MOF, thymol, encapsulation, controlled release, cherry tomatoes

## Abstract

Thymol is a phenol monoterpene with potential antifungal, antioxidant and antibacterial activities. Due to the low water solubility and high volatility of thymol, encapsulation serves as an effective tool during application. In the present study, cyclodextrin (CD)-based metal-organic-frameworks (MOFs) were synthesized using α-CD, β-CD, and γ-CD as organic building blocks, and further complexed with thymol to produce three CD-MOF-THY inclusion complexes (ICs). The encapsulation content, release kinetics and fruit preservation effect of ICs were analyzed. Results showed that thymol was well embedded in γ-CD-MOFs, with the highest encapsulation content of 286.7 ± 8.4 mg/g. Release kinetics revealed that CD-MOFs exhibited a controlled release effect toward thymol for 35 days. The release kinetics of three ICs fit the Rigter–Peppas model well, with γ-CD-MOF-THY showing the lowest release rate constant of 2.85 at 50 °C, RH 75%. Moreover, γ-CD-MOF-THY exhibited a remarkable preservation performance on cherry tomatoes with the lowest decay index (18.75%) and weight loss (5.17%) after 15 days of storage, suggesting this material as a potential fresh-keeping material for fruit and vegetable preservation.

## 1. Introduction

Thymol, derived from *Thymus vulgaris* (common thyme), ajwain, and various other plants, is a phenol monoterpene with an aromatic structure and found to exhibit potential antifungal, antioxidant and antibacterial activities. Due to its low water solubility (1.0 g/L) and high volatility, thymol is often encapsulated for application. Various wall materials such as cyclodextrin (CD), lipid, glycerin, and soy protein isolate have been employed in the encapsulation of thymol. The final products are usually presented the form of hydrogel [1,2,3], liposome [4,5], nanoparticle [6,7,8,9], microcapsule [10], inclusion complex [11,12,13,14,15], building with spray drying [10,16,17], freeze drying [18], electrospinning [19,20], kneading [21,22] and other methods. Pivetta [23] determined the anti-inflammatory properties of thymol in nanostructured lipid carriers consisting of Illipe butter and Calendula oil, and the group treated with thymol-encapsulated liposomes significantly increased the in vitro wound healing capability compared with the blank group. Further animal model experiments demonstrated the positive effect of thymol-encapsulated liposomes on wound healing. Piombino [24] prepared thymol-encapsulated microcapsules by lignin through a sustainable ultrasonication method, achieving an encapsulation efficiency of 51%. Thereafter, thymol was slowly released from the matrix due to its inherent lipophilicity. Wu [25] loaded thymol into zinc metal–organic framework (MOF); the thermogravimetric analysis demonstrated a loading rate of 3.96%, and thymol loaded in Zn@MOF exhibited excellent antibacterial activity against a three-strain cocktail of *E. coli* O157:H7. Koosehgol et al. [3] fabricated an antibacterial composite film based on chitosan/polyethylene glycol fumarate/thymol as wound dressings, and films with thymol content of 1.8% showed great antibacterial activity against both Gram-positive and Gram-negative bacteria. Additionally, Chen [26] obtained a more stable complex by electrospinning cellulose acetate with β-CD. The combination of encapsulated wall materials ensured sustainable thymol release with good cytocompatibility, with an encapsulation content of 48.6 mg/g. However, the encapsulation efficiency and loading capacity of thymol are generally insufficient. Furthermore, the wall materials should be biocompatible and renewable in the application process of the food industry. Hence, the preparation procedure of thymol ICs warrants further improvements to fulfill the requirements of food applications.

MOF is a sort of crystalline material with a highly porous structure and adjustable porosity, which is usually self-assembled by alkaline metal clusters and organic linkers under certain conditions and has a periodic network arrangement [27]. Recently, MOFs and their derivates have attracted much attention due to their antibacterial activity. Notably, MOFs based on CDs are green materials with a high specific surface area. Significantly, the combination of CDs and MOFs could potentially provide a promising encapsulation effect towards thymol.

The natural CDs mainly include α-CD, β-CD and γ-CD, which are composed of 6, 7, and 8 D-glucopyranose units. With the increase of glucopyranose units, the cavity sizes of α-CD, β-CD, and γ-CD increase with a corresponding cavity diameter of 0.57, 0.78, and 0.95 nm [28]. Natural CDs and guest molecules generally form complexes with stoichiometric ratios of 1:1, 1:2, 2:1, or 2:2, which largely limit the encapsulation efficiency. After the coordination of CDs with alkali metals, the pore distribution in the crystals significantly increases, leading to an improved CD-MOF encapsulation capacity. Li et al. [29] synthesized α-CD-MOF-Na and α-CD-MOF-K to encapsulate ethylene, and it turned out that the encapsulation content of the two CD-MOFs was much higher than that of α-CD and V-type starch. In addition, a higher menthol encapsulated capacity was obtained in β-CD-MOF than pristine β-CD, and menthol/β-CD-MOF-IC is more tolerant to high temperatures [30]. Moreover, compared with γ-CD, the thymol encapsulation capacity of γ-CD-MOFs generated by potassium salts was increased by two- to three-fold. It was considered that anions in the preparation solution led to different three-dimensional CD-MOF structures by affecting the symmetry relationship between ions and atoms, thus resulting in different thymol encapsulation contents [31]. Further investigations on release kinetics also revealed that β-CD-MOF provided menthol with better controlled-release behaviors [30]. As for the CD-MOFs prepared by three types of cyclodextrins, their differences in thymol encapsulation ability and release behavior are currently unrevealed.

Thymol showed an excellent antibacterial effect on a variety of pathogens [32,33]. It is of great significance to employ thymol in preserving fruits and vegetables. The encapsulated thymol in CD-MOFs may achieve a great controlled-release behavior, thus prolonging the shelf life of fruits and vegetables. In this study, thymol was encapsulated in three kinds of CD-MOFs prepared by a vapor diffusion strategy. Investigations of thymol’s preservation effect on cherry tomatoes were carried out to provide new scenarios for thymol application in food industry.

## 2. Materials and Methods

### 2.1. Materials

Cherry tomatoes (*Solanum lycopersicum var. cerasiforme*) were purchased from Guangzhou Agricultural Products Market, and fruits with no mechanical damage and good maturity were selected. The fruit averagely weighted 14.97 g. Thymol (CAS number 89-83-8, 99%) and γ-CD (98%) were provided by Aladdin Biochemical Technology Co., Ltd. (Shanghai, China). α-CD and β-CD were obtained from Macklin Biochemical Co., Ltd. (Shanghai, China). All other chemical reagents were of analytical grade and purchased from local suppliers.

### 2.2. Preparation of CD-MOFs

The CD-MOFs in this study were prepared according to a previous research [34]. Briefly, 8 mmol of KOH and 1 mmol of CD were weighted and dissolved in 20 mL of deionized water. The mixture was then kept stirring at room temperature for 3 h to obtain the CD concentration of 0.05 mmol/L. The final solution was then filtered by a 0.45 μm organic filter membrane and placed in a closed environment at 25 °C, so that the diffusion medium (anhydrous ethanol) could naturally diffuse into the solution, and white crystal products were obtained after several days. The collected white crystals were successively washed with absolute ethanol for three times, emerged in dichloromethane for 48 h and finally vacuum-dried at 40 °C for 24 h. Crystals were named α-CD-MOF, β-CD-MOF, and γ-CD-MOF according to the type of CD used.

### 2.3. Encapsulation of Thymol into CD-MOFs

The thymol-loaded ICs were prepared by a high-temperature adsorption method described in a previous study [35]. Briefly, CD-MOFs and thymol were homogeneously mixed at a mass ratio of 3:7 (CDs: thymol) in a hydrothermal synthesis reactor and further heated at 80 °C for 2 h. After the mixture was cooled to room temperature, the excess thymol was removed by washing the mixture three times with absolute ethanol. The obtained pellet was centrifuged and filtered, and finally vacuum-dried at 40 °C for 24 h. The final ICs were in powder status and named α-CD-MOF-THY, β-CD-MOF-THY and γ-CD-MOF-THY.

### 2.4. Quantification of Thymol in ICs

The thymol encapsulation content of ICs was determined by gas chromatography (GC) [36]. ICs (approximately 10 mg) were accurately weighted and dissolved in 1 mL of deionized water. Then, 2 mL of ethyl acetate was added to extract thymol. The supernatant was collected and further analyzed by a GC system (7090B, Agilent Technologies, Santa Clara, CA, USA). A silica HP-5 capillary column (30 m × 0.25 mm × 0.25 μm; Agilent Technologies) and a flame ionization detector were employed for thymol content quantification. The carrier gas was nitrogen, and the flow rate was set at 1 mL min^−1^ with a split ratio of 1:15. Both the inlet and detector temperatures were 250 °C. The temperature was set as follows: column temperature was firstly held at 80 °C for 1 min and then increased to 250 °C at a rate of 30 °C min^−1^ with a final hold of 2 min. Quantification of thymol was calculated in terms of the relative peak area of each sample on the thymol standard curve (R^2^ > 0.99), and the thymol encapsulation content of ICs was calculated using the following formula:(1)$$\mathrm{Encapsulation}\;\mathrm{content}\;\left(\mathrm{EC},\frac{\mathrm{mg}}{g}\right)=\frac{\mathrm{weight}\;\mathrm{of}\;\mathrm{thymol}\;\mathrm{in}\;\mathrm{ICs}}{\mathrm{weight}\;\mathrm{of}\;\mathrm{ICs}}\;$$

### 2.5. Release Kinetics

The release kinetics of thymol from the ICs were determined under different temperatures and relative humidity (RH) conditions. Approximately 0.5 g of ICs and pure thymol (as a control) were placed in a glass plate at 4, 25, or 50 °C under an RH of 50% or 75%. Samples were acquired at different intervals to quantify the release content of thymol by GC., and the release ratio was calculated using the following formula:(2)$$\mathrm{Release}\;\mathrm{ratio}\;\left(\%\right)=\frac{M_{\infty }-{M_{t}}^{\prime }}{M_{\infty }}\;$$
where ${M_{t}}^{\prime }$ denotes the thymol content in ICs at time *t* and $M_{\infty }$ denotes the original thymol content in ICs.

The Higuchi model (Equation (3)) and Rigter–Peppas model (Equation (4)) were employed to describe the release kinetics of thymol:(3)$$M_{t}=K_{H}\sqrt{t}\;$$
(4)$$kt^{n}=\frac{M_{t}}{M_{\infty }}\;$$
where $K_{H}$ denotes the dissolution constant (concentration per time^1/2^), $\frac{M_{t}}{M_{\infty }}$ denotes the release fraction of thymol at time t, *k* denotes the release velocity constant and n stands for the release exponent [37].

### 2.6. Preservation Effect of Thymol ICs on Cherry Tomatoes at Room Temperature

Fresh cherry tomatoes with no physical damage were sorted to choose full red ones, and approximately 300 g of cherry tomatoes were used in each group. The experiment was carried out at 25 °C with RH of 55% ± 5% in sealed boxes (4 L) and lasted for 15 days. Samples were taken out for further determinations every three days.

The hardness was evaluated by a fruit hardness tester (Shunkeda Technology Co., Ltd., Beijing, China). After removing the skin from a single fruit, three points were selected on the surface of cherry tomatoes. The fruit hardness tester with zero adjustment and correction was pressed to the pulp with uniform force in the vertical direction, and the results were recorded. And each group of samples was measured three times.

Approximately 30 g of cherry tomatoes was weighed. The pulp juice was extracted and dipped on the surface of the light-blocking prism of a handheld saccharometer (Pxtong Technology Co., Ltd., Guangzhou, China). Results were recorded as the soluble solids content.

The decay index and weight loss were calculated according to the following equation [38]:(5)$$\mathrm{Deacy}\;\mathrm{index}\;\left(\mathrm{DI}\right)=\frac{\sum \left(\mathrm{DI}\;\mathrm{rating}\;\times \;\mathrm{no}.\;\mathrm{of}\;\mathrm{cherry}\;\mathrm{tomatoes}\;\mathrm{at}\;\mathrm{DI}\;\mathrm{rating}\;\mathrm{level}\right)}{\mathrm{Total}\;\mathrm{number}\;\mathrm{of}\;\mathrm{treated}\;\mathrm{cherry}\;\mathrm{tomatoes}}\;$$
where 0: no decay; 1: decay extent ≤ 25%; 2: 25% < decayed extent ≤ 50%; 3: 50% < decayed extent ≤ 75%; 4: 75% < decayed extent.
(6)$$\mathrm{Weight}\;\mathrm{loss}\;\left(\%\right)=\frac{\mathrm{Wi}-\mathrm{Wf}}{\mathrm{Wi}}\times 100\%\;$$

And groups were established as followed:(1)α-CD-MOF-THY group: 30 cherry tomatoes treated with 50 mg of α-CD-MOF-THY powder placed in the four corners of the sealed box.(2)β-CD-MOF-THY group: 30 cherry tomatoes treated with 50 mg of β-CD-MOF-THY powder placed in the four corners of the sealed box.(3)γ-CD-MOF-THY group: 30 cherry tomatoes treated with 50 mg of γ-CD-MOF-THY powder placed in the four corners of the sealed box.(4)Thymol group: 30 cherry tomatoes treated with 50 mg of solid thymol powder placed in the four corners of the sealed box.(5)Control group: 30 cherry tomatoes with no treatment in the sealed box.

### 2.7. Statistical Analysis

Samples were analyzed in triplicate and presented as the mean ± standard deviation (SD). Differences between groups were analyzed by one-way analysis of variance at the 95% significance level (*p* < 0.05) using SPSS 19.0 software (IBM Corp., Armonk, NY, USA). For preservation experiments, presented data points are the means of three replications without a significance test.

## 3. Results and Discussion

### 3.1. Encapsulation Contents in ICs

The thymol encapsulation content of three ICs is shown in Figure 1. The γ-CD-MOF showed the highest thymol-encapsulated content of 286.7 ± 8.4 mg/g, followed by β-CD-MOF and α-CD-MOF, for which values were also higher than those of previously reported wall materials [3,25,26,36,39,40]. Presumably, γ-CD-MOF is more suitable for thymol encapsulation due to its matched cavity diameter with the dimension of this aroma molecule. Our previous study has found that γ-CD-MOF (prepared by KOH) showed regular cubic crystalline structures with smooth and particle surfaces [31]. Generally, the hydrophobic cavity of cyclodextrin and newly formed pores between cyclodextrin molecules are two types of cavity structures that exist in CD-MOF crystals. According to the site-preference mechanism, thymol tends to first enter the cyclodextrin cavity during high-temperature adsorption, then partial excess thymol begins to enter the newly formed intermolecular cavities. Since an efficient combination between the host and guest requires geometrical fitting [41], the significant difference in encapsulation capacity may be due to the difference in cavity size distribution among the three kinds of CD-MOFs. The cavity size of α-, β-, and γ-CDs increases gradually, and the newly formed cavity size is closely related to the length of the K−O bond and the arrangement of CD molecules [42]. Studies have shown that the monoclinic space group crystallizes α-CD-MOF and β-CD-MOF, and the diameter of the intermolecular cavity is about 0.5 nm [29,42], while γ-CD-MOF is crystallized in a cubic space group with high symmetry [31], and cavities are found in the structure with diameters of 1.7 nm, 1.0 nm, and 0.4 nm [31,41]. Thymol binds to CD-MOF through hydrophobic interactions and van der Waals forces [30]. CDs with larger cavities provide the aroma with enough space to enter in, which increases the binding probability of thymol combining with CD-MOF [31]. In contrast, the smaller cavities of CDs hinder thymol molecules from entering the hydrophobic cavity. Therefore, thymol has the highest binding efficiency with γ-CD-MOF. Hu et al. [35] prepared three CD-MOFs by incorporating potassium nitrate with CDs in an alkaline aqueous solution, and the menthol encapsulation capacity of β-CD-MOF was higher than the other two CD-MOFs. It was explained that the cavity diameter of α-CD was inadequate, and the γ-CD was excessive to host menthol effectively. Thus, generating better-encapsulated capacity requires tight steric complementarity between host and guest.

### 3.2. Release Kinetics

As is shown in Figure 2, thymol encapsulated in γ-CD-MOF showed a slower controlled release behavior than the other groups in all conditions. After being stored at 4 °C with a RH of 50% for 35 days (Figure 2a), only 34.44% of thymol was released from α-CD-MOF, and γ-CD-MOF-THY showed a minimum thymol release content of 18.01%. Hence, CD-MOFs could serve as an effective tool to achieve the sustainable release of thymol. Obviously, temperature could accelerate the release of thymol (Figure 2a–c). Under high temperature, all unencapsulated thymol vaporized within 5 days (Figure 2c). However, the γ-CD-MOF-THY still showed the lowest release rate compared with the other two counterparts. Kayaci et al. [43] once prepared three kinds of vanillin/CD ICs, and results showed that the interaction strength between vanillin and the CD cavity was ordered as γ-CD > β-CD > α-CD. They believed that the bigger cavity size of γ-CD resulted in a better fit and size match between the vanillin molecule and the γ-CD cavity. We assumed that the lowest release rate of γ-CD-MOF-THY could be due to the stronger non-polar bonds existing in γ-CD-MOF-THY. Despite the larger size of the hydrophobic cavity in γ-CD, thymol molecules can be allowed to move in the cavity without reaching the hydrophilic surface. Hence, a large driving force, such as high temperature or high humidity is required to accelerate molecular motion, further destroying the ordered structure of MOF.

Moreover, with the RH increasing from 50% to 75% at 25 °C (Figure 2b,d), the thymol release content from α-CD-MOF-THY and β-CD-MOF-THY increased. In contrast, the release content of thymol from γ-CD-MOF-THY did not change significantly. However, thymol without encapsulation showed fast and continuous release with a linear behavior for 35 days (Figure 2b). The complex structure of γ-CD-MOF-THY was more stable than α-CD-MOF-THY and β-CD-MOF-THY. A similar phenomenon was observed when the temperature was set as 50 °C (Figure 2c,e). The structure of the CD-MOFs disintegrated gradually under high humidity, which decreased the IC stability and significantly accelerated the release rate of thymol [44]. Based on previous research [30], the solubilization of the wall material could directly lead to a significant loss of thymol, which was closely correlated with the concentration of water molecules surrounding ICs.

To better describe the release kinetics of thymol from three CD-MOFs, we applied two mathematical models to simulate the experimental data (Table 1). The Higuchi model best describes the release kinetics of encapsulated guest molecules of low solubility encapsulated in solid or semisolid matrices, and the Rigter–Peppas model is applicable when more than one release mechanism is involved [37].The simulations showed that the Rigter–Peppas model with higher correlation coefficients ($R_{adj}^{2}$) presented more preferable fits than those of the Higuchi model. The release rate constant *k*, also known as the release velocity constant, is quite related to the structural changes and geometry of the encapsulation system.

As is shown in Table 1, the release rate constant increased with temperature and humidity. The *k* of γ-CD-MOF-THY was much smaller than that of α-CD-MOF-THY and β-CD-MOF-THY at 50 °C. At room temperature, the release rate constant could be ordered in the sequence of *k*_β-CD-MOF-THY_ > *k*_α-CD-MOF-THY_ > *k*_γ-CD-MOF-THY_. Moreover, the n value of γ-CD-MOF-THY is even lower than 0.45, which indicates that thymol is released from γ-CD-MOF due to the concentration gradient [45].

### 3.3. Preservation Effect of Thymol ICs on Cherry Tomatoes

Cherry tomatoes are susceptible to transportation and storage due to pathogen infection and mechanical damage. Therefore, decay inhibition is of great importance in the preservation process of cherry tomatoes. Changes in the decay indices of cherry tomatoes treated by different thymol ICs are shown in Figure 3a. Thymol ICs in each group showed obvious preservative and fresh-keeping effects during 15 days of storage at room temperature for cherry tomatoes. After 15 days, cherry tomatoes in the control group showed the highest decay index of 67.5%. γ-CD-MOF-THY showed the lowest decay index of 18.75%, which was same as the thymol group, followed by α-CD-MOF-THY (33.3%) and β-CD-MOF-THY (48.75%). To visually observe the preservation effect of γ-CD-MOF-THY, the appearance of cherry tomatoes treated in the control group, thymol group, and γ-CD-MOF-THY group are shown in Figure 3e as well. Fewer bacterial colonies were found in the γ-CD-MOF-THY group compared with the control group. Thus, it could be concluded that the fresh-keeping ability is stronger with increased thymol content in ICs.

Soluble solids generally include soluble sugar, tannic acids, and other water-soluble substances, mainly reflecting the fruit sugar content. The soluble solid content of cherry tomatoes is closely related to fruit quality, processing characteristics, and fruit physiological activities, and a certain amount of sugar can make the fruit tastier. From Figure 3b, it could be observed that the soluble solid content of cherry tomatoes increased in the early stage, decreased in the middle stage, and occasionally increased in the later stage, but the overall change scale was insignificant. On the one hand, the reason may be due to the individual differences between cherry tomatoes. On the other hand, cherry tomatoes usually consume much energy to maintain basic physiological activities in the early storage stage, and the hydrolysis rate of starch in cherry tomatoes is higher than that of soluble sugar consumption by respiration, leading to the accumulation of soluble solid content [46]. In the middle and late stages, cherry tomatoes would mature gradually until aging. The hydrolysis rate of starch decreases at this stage, and the consumption rate of soluble sugar is higher than the production rate, leading to a decreased soluble solids content. The mass reproduction of microorganisms in the later stage may consume the soluble sugar in cherry tomatoes, resulting in the decreasing trend of soluble solid content. Notably, cherry tomatoes treated in the γ-CD-MOF-THY group had the highest soluble solid content compared with other groups after 15 days of storage at room temperature. Hence, γ-CD-MOF-THY had a better preservative effect to alleviate the soluble solid loss of cherry tomatoes and exhibited a great potential to prolong the shelf life of the whole fruit storage period.

The loss of fruit mass during storage is partly induced by respiratory consumption, and the weight loss rate is an essential index to evaluate fruit quality. It can be learned from Figure 3c that the weight loss of cherry tomatoes treated in different IC groups increased with prolonged storage time. The fruit in the γ-CD-MOF-THY group showed the lowest weight loss of 5.17% after 15 days, which was related to the higher thymol content within γ-CD-MOF-THY, leading to a better inhibition effect on the metabolic level of cherry tomatoes. In addition, hardness is another crucial index modulating fruit taste, thus affecting consumers’ acceptability and selectivity. Fruit softening occurs due to deterioration in the cell structure, cell wall composition, and intracellular components [47]. As depicted in Figure 3d, the hardness of cherry tomatoes showed a downward trend with the extension of storage time, ranging from 2.29 to 0.65 kgf/cm^2^. Therefore, α-CD-MOF-ICS may have the potential of maintaining fruit hardness.

## 4. Conclusions

The MOF materials prepared based on CDs have a good encapsulation effect for thymol. γ-CD-MOF exhibits better complex ability with thymol compared with α-CD-MOF and β-CD-MOF, with the highest encapsulation content of 286.7 ± 8.4 mg/g. The release kinetics revealed that thymol encapsulated in γ-CD-MOF was sustainably released for 35 days, which was also well fitted by the RigterPeppas model. The better-controlled release effect of γ-CD-MOF-THY further lays the foundation for the IC to extend the shelf life of cherry tomatoes. The decay index of whole cherry tomatoes treated with γ-CD-MOF-THY decreased from 67.5% (control group) to less than 20% during storage at room temperature for 15 days, which was similar to the preservation effect of free thymol (18.75%). The preservation effect was stronger with the increase of thymol content in ICs, and γ-CD-MOF-THY showed a better preservation effect towards cherry tomatoes. Findings in this study have theoretical guiding significance for fruit and vegetable preservation.

## Figures and Tables

**Figure 1 foods-11-03818-f001:**
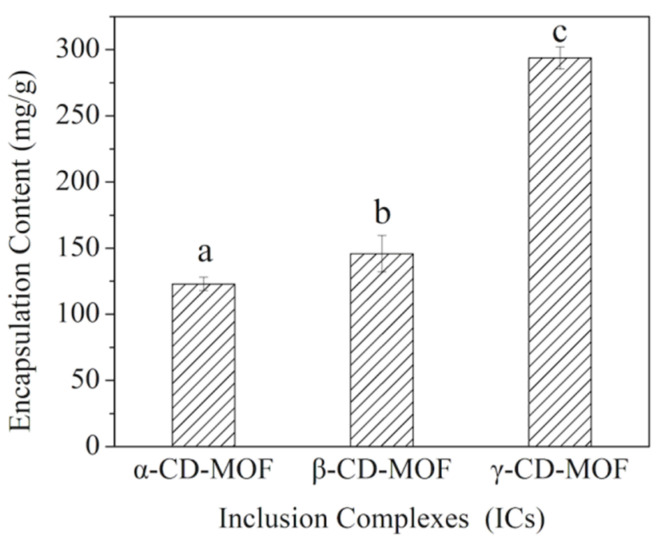
Encapsulation content of thymol in inclusion complexes. Different superscript letters (a–c) represent statistically significant differences (*p* < 0.05).

**Figure 2 foods-11-03818-f002:**
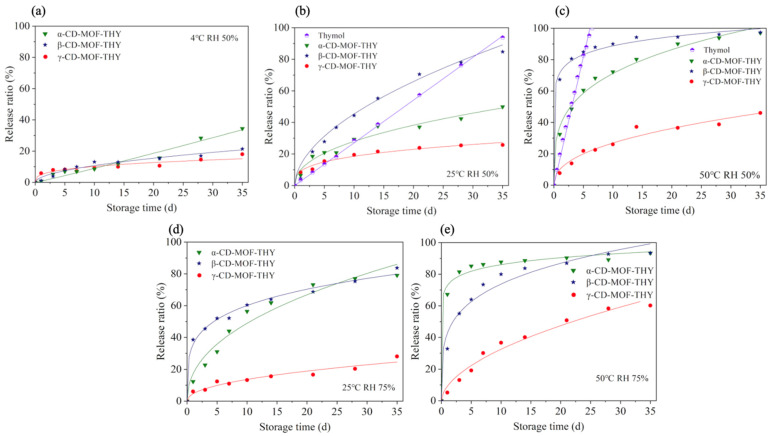
In vitro kinetics of thymol released from different γ-CD-MOFs at different temperatures and RH values stored for 0 to 35 days: (**a**) 4 °C, RH 50%, (**b**) 25 °C, RH 50%, (**c**) 25 °C, RH 50%, (**d**) 50 °C, RH 75% and (**e**) 50 °C, RH 75%. *p* < 0.05.

**Figure 3 foods-11-03818-f003:**
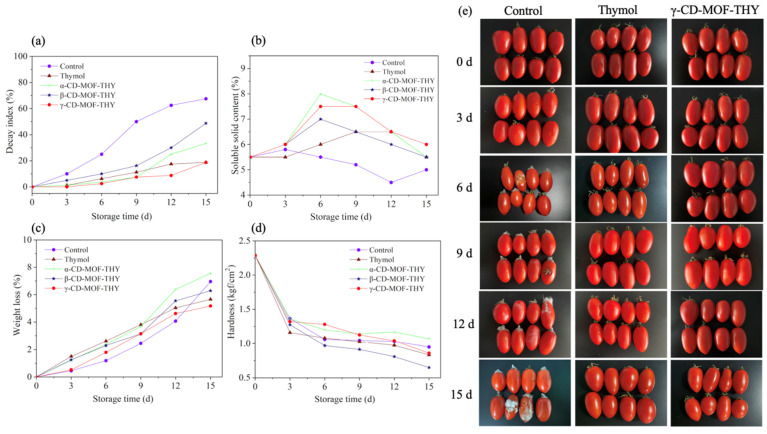
Decay index (**a**), soluble content (**b**), weight loss (**c**) and hardness (**d**) change of the whole cherry tomatoes treated in different groups of thymol ICs. Appearance (**e**) of the whole cherry tomatoes treated in the control group, thymol group and γ-CD-MOF-THY group.

**Table 1 foods-11-03818-t001:** Kinetic constants and correlation coefficients of inclusion complexes release based on different models.

ICs	Temperature, RH	Higuchi Model		Rigter-Peppas Model		
		$R_{adj}^{2}$	*k*	$R_{adj}^{2}$	*k*	n
α-CD-MOF-THY	4 °C, 50%	0.93675	1.74746	0.95935	0.88807	1.05567
β-CD-MOF-THY		0.92291	1.86772	0.94674	1.78816	0.52674
γ-CD-MOF-THY		0.18290	1.66846	0.81410	2.22600	0.31403
α-CD-MOF-THY	25 °C, 50%	0.95562	3.14312	0.96435	3.15379	0.44954
β-CD-MOF-THY		0.92655	3.95740	0.95932	3.62073	0.53908
γ-CD-MOF-THY		0.87004	2.45716	0.97063	3.01640	0.30902
α-CD-MOF-THY	25 °C, 75%	0.97401	4.01374	0.95702	4.15923	0.45127
β-CD-MOF-THY		0.64462	4.27883	0.98893	5.96748	0.22721
γ-CD-MOF-THY		0.89675	2.03278	0.92606	2.15897	0.46771
α-CD-MOF-THY	50 °C, 50%	0.83865	5.03135	0.98823	6.11501	0.27956
β-CD-MOF-THY		0.25615	5.36544	0.99253	8.46514	0.09305
γ-CD-MOF-THY		0.95843	2.94662	0.96009	3.16589	0.42854
α-CD-MOF-THY	50 °C, 75%	0.15271	5.24209	0.98064	8.51740	0.09021
β-CD-MOF-THY		0.82193	5.01330	0.84759	6.70909	0.26986
γ-CD-MOF-THY		0.91117	3.45773	0.92958	2.84530	0.66918

Note: $R_{adj}^{2}$ is the correlation coefficient. *k* is the release rate constant. n is the release exponent.

## Data Availability

The data are available from the corresponding author.

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
