# Peer review of "Controlled Release of Thymol by Cyclodextrin Metal-Organic Frameworks for Preservation of Cherry Tomatoes"

_foods, 2022, doi:10.3390/foods11233818_

Round 1

Reviewer 1 Report

Comments:

1. Statistical analysis to state the number of replicates

2. Line 181-change the word 'tends first to' to 'tends to first'

Author Response

  1. Statistical analysis to state the number of replicates

Thanks! We have stated the number of replicates in the context. (Line213-271)

  1. Line 181-change the word 'tends first to' to 'tends to first'

Thank you for the reviewer’s careful work! We have revised the term into “tends first to”.

Reviewer 2 Report

This MS in the topic "Controlled Release of Thymol by Cyclodextrin Metal-Organic Frameworks for Preservation of Cherry Tomatoes" has a question at some point. The author needs to clarify and revise as follows.

M&M: This MS was not clear in the method to be used. The author needed more explanation step by step.

- Line 117: After preparation of the finished product, what is the status of -CD-MOF-THY, -CD-MOF-THY and -CD-MOF-THY? in solid, semi-solid, or dust? Please explain more. Or explain more in the Result.

- Line 127-130: It is still unclear how to calculate the content of thymol from the standard curve of GC and the content of thymol in ICs. The author needs to explain step by step. and clarified the formula of "encapsulation capacity" that is related to "encapsulation content" in part of the result. Which one does the author like to use?

- Line 132-136: It is still unclear about the method of measuring kinetic release related to the result (Figure 2) in terms of "Relative ratio (%)". That is still not understood in terms of "Ratio" but reported in units of "percentage". The author needs to explain more and show the reference of this method.

- Line 145-166: It is still unclear about the method. This section cannot be accepted as is; it must be revised for method. 

How to use CD-MOF-THY, -CD-MOF-THY and -CD-MOF-THY to treat cherry tomatoes? Drip in ICs solution, spray, or use drip in solid dust. (Line157-165)

How to measure soluble content?

How to measure Hardness?

And the important indicator for this MS must be to measure the bacterial count from tomato fruit that shows the direct reaction of Thymol.

Statistical analysis: It is still unclear. Which statistic parameter did the author calculate? Result in Figure 1? And for other result as in Figure 2 and 3. Do authors use statistical analysis?

Result:

Figure 1:  The result and M&M were not related together. such as the Author label Y axis with "Encapsulation Content (mg/g)" but in M&M line 130, shown calculated in terms of "Encapsulation capacity". How are they related together?

Figure 2: The result and M&M were not related together.

Line 225: in topic of “Preservation effect of thymol ICs on cherry tomatoes” have to discussion with the direct reaction of Thymol to antibacterial. And other parameters need statistical analysis to show a good result compared with control.

Reviewer 3 Report

1. Please describe in detail the variety (scientific name) of Cherry tomatoes, delivery and preservation methods, maturity, and average weight.

2. Fig1 Pvalue?

3. Fig3 Pvalue?

4. Is there any microbiological analysis?

5. It is recommended to provide SEM images of different CD microcapsules

Round 2

Reviewer 1 Report

All corrections are addressed. 

Reviewer 2 Report

This reversion manuscript can be accept.